# The influence of horizontal violence on intention to leave among Jordanian nurses: A cross-sectional study

**Ahmad H. Al-Nawafleh**[1]*, **Zaid M. Al-Hamdan**[2], **Hamzah Bawayzah**[2], **Hala Bawadi**[3]

**1** Faculty of Nursing, Mutah University, Mu'tah, Jordan, **2** Faculty of Nursing, Jordan University of Science and Technology, Irbid, Jordan, **3** School of Nursing, The University of Jordan, Amman, Jordan

* alnawafleh@mutah.edu.jo

**Data Availability Statement:** All relevant data are within the manuscript.

**Funding:** The author(s) received no specific funding for this work.

## Abstract

### Aim

To determine nurses' perception of horizontal violence and its relationship with intention to leave among Jordanian nurses.

### Background

Horizontal violence is detrimental to healthcare organizations. Healthcare employees who are victims of horizontal violence may become unable to perform well, living with severe stress. Few studies in the world and no study in Jordan has examined the influence of horizontal violence on intent to stay among nurses.

### Method

A descriptive, cross-sectional, correlational study design was used in this study.

### Result

520 registered nurses were recruited, and 436 surveys were returned, resulting in an 83% response rate. The findings showed that, horizontal violence is moderate among nurses in Jordan. The mean score for nurse's intent to leave work indicated a solid willingness to leave the job. The correlation analysis revealed that there was a significant relationship between horizontal violence and intent to leave. Female nurses reported significantly higher scores than males; also, there is a significant difference in responses in horizontal violence based on educational level.

### Conclusion

In this study, it was found that horizontal violence in Jordanian organizations is moderate, and it has a significant relationship with nurses' intention to leave.

**Competing interests:** The authors have declared that no competing interests exist.

## Implication for nursing management

Nursing managers have a vital role in ensuring and enforcing the 'zero violence' policy in their department, which means that no reported incident of horizontal violence will be ignored. Each identified case of horizontal violence will be investigated and addressed appropriately.

## 1. Background

The nursing work environment is complex, and among the diversity of factors affecting nurses' job performance requirements and expectations, relationships within the work environment stand as a significant factor. The workplace environment involves building relationships with colleagues, other healthcare providers, patients, and their families, and visitors [1]. Nonetheless, some of these relationships might be unhealthy and stressful, especially when someone or a group has domination over other colleagues. Horizontal violence is a term that is used to describe a specific type of violence between employees, including physical assaults, expressions of severe anger, conflicts between individuals, and psychological destructive behaviors such as bullying and intimidation [2]. Horizontal violence was used in literature alternately with workplace bullying, incivility, or lateral violence [3].

Horizontal violence is detrimental to healthcare organizations. Healthcare employees who are victims of horizontal violence may become unable to perform well, living with severe stress, and suffering from negative emotions toward attending to work and to their colleagues [2, 4–6].Various factors contribute to horizontal violence practices. These factors may be categorized into personal traits and organizational factors. Some of the personality traits of the perpetrator are shown through frequent criticizing, insulting, putting victim under tremendous pressure, speaking in a loud voice or shouting, blaming, humiliating, and physical annoyance [7–9]. Nurses could experience work-related horizontal violence from diverse backgrounds and at any level. Some horizontal violence examples involve invalid information being exchanged, holding responsibilities for co-workers' duties, getting scolded in front of others, no acknowledgment of work, or confrontation of employees in front of others. Perceptions of horizontal violence were found to be connected with the intent to leave [10].

A descriptive correlational study was conducted by Armmer and Ball (2015) to assess horizontal violence among nurses and determine the presence of a relationship between horizontal violence and intent to leave. A total of 300 participants were randomly selected, results demonstrated that horizontal violence was experienced across different groups of ages and experiences. This violence included failure to be acknowledged in front of others, reprimanded or confronted in front of others, untrue information about the nurse being passed or exchanged, and being held responsible for co-workers' duties.

Correlational analyses showed that novice, younger nurses had higher intention to leave due to horizontal violence perpetrated by older nurses who had more experience. Nurses who have worked longer in the organization were more likely to have experienced some form of horizontal violence (i.e., horizontal violence has been integrated within the organization). Nurses of horizontal violence had higher intention to leave (Armmer and Ball 2015).

In a cross-sectional study conducted in Lebanon by [4], the aim was to identify the occupational violence prevalence, characteristics, consequences, and associated factors among nurses. A total of 915 were selected through random sampling to respond to mailed questionnaires. Questionnaires included demographic background, exposure to violence, and burnout. Results

showed that less than two thirds (62%) witnessed verbal violence, while 10% experienced physical abuse. Moreover, less than a third (31.7%) indicated their willingness to leave their job, and 22.3% were not sure. High levels of emotional exhaustion were reported by more than half (54.1%), while 28.8% suffered from depersonalization. Evidence in literature showed that sociodemographic characteristics such as gender and work shift correlated with the perceived level of horizontal violence among nurses [4]. In Jordan, nurses perceived exposure to horizontal violence was found to be linked with their level of competence; nurses who had higher level of competence reported being less victimized by their colleagues [11]. However, no study has assessed the relationship between horizontal violence and intent to leave among nurses working in Jordanian hospitals. Therefore, this study aimed to determine nurses' perception of horizontal violence and its relationship with the intention to leave among Jordanian nurses. This study attempting to answer the following questions:

1. What is the current level of horizontal violence among Jordanian nurses?

2. Is there a relationship between horizontal violence and intent to leave work?

3. Are there differences in horizontal violence based on sociodemographic variables including Age, gender, marital status, years of experience, and educational level?

## 2. Methods

### 2.1 Research design and participants

This study adopts a descriptive, cross-sectional, and correlational design. The study recruited a convenience sample of nurses based on the total nursing population count in the selected hospitals. The sample size was determined using G Power 3.1, given an alpha level of 0.05, the medium effect size of 0.5, and the desired statistical power level of 0.8, a total population size of 2000 nurses from the 4 participating hospitals; the minimum required sample size is 323. According to Fincham (2008), a target response rate of 60% should be the goal for any survey-based research [12]. Therefore, surveys distributed for 520 nurses working in the 4 participating hospitals to achieve the required sample size of 323. Inclusion criteria were: (a) nurses working in hospitals with different classifications: nurse aid, practical nurse, registered nurses; (b) nurses working as full-time employees.

### 2.2 Measures

Horizontal violence was measured using the Negative Acts Questionnaire- revised (NAQ-R) developed by Einarsen et al. in 2009; the questionnaire was answered using a five-point Likert scale (Never, Now, and then, Monthly, Weekly, and Daily). The respondents answered for a look-back period of 6 months to assess the frequency of exposure to any of the 22 negative behaviors. The value of 1 means that the respondent has never experienced negative behavior, while the value of 5 means that the respondent is experiencing negative behavior every day. The total score of the Likert-type items, in addition to the extra question, can be calculated as a total sum, with higher values indicating a higher frequency of exposure to negative acts. Employees with a score lower than 33 did not experience horizontal violence. Employees with a score between 33 and 45 may be considered as experiencing horizontal violence occasionally, and employees who score above 45 can be considered victims of horizontal violence [13]. The NAQ-R has strong reliability and constructs validity [14] tested the psychometric characteristics of the questionnaire; the NAQ-R yielded a high level of internal consistency (Cronbach's alpha = 0.90) with all factor saturation scores exceeding 0.70. The factors under the scale are highly correlated and confirm the satisfactory level of construct validity, which was assessed

using psychophysical health measures, work satisfaction, efficacy, dedication to the organization, and similar.

Intention to leave was measured using anticipated turnover scale (ATS), which was developed by [15] to determine intent to leave a nursing position or anticipated turnover in nursing. The survey consists of 12 items and is intended to measure the intent to leave on a 7-point Likert scale ranging as follows: agree strongly (AS), moderately agree (MA), slightly agree (SA), uncertain (U), slightly disagree (SD), moderately disagree (MD), and disagree strongly (DS). Items are either positively or negatively stated; positively stated questions are 2, 3, 4, 5, 9, 11, and 12, and they are scored on a scale of 7 to 1, with AS being 7 points and DS being 1 point. On the other hand, while 1, 6, 7, 8, and 10 are negatively stated questions and are scored from 1 to 7 points, with AS being equal to 1 point and DS equal to 7. Negatively stated questions were reversed prior to data analysis. An average score of 3.5 and above means a strong willingness to leave the job, while below 3.5 indicates a weak willingness to leave the job [15, 16]. Similar to NAQ, the translation-back-translation procedure was followed, and then the tool was subjected to face validity by 3 experts. All comments and feedback provided about the tool items were taken into consideration and modified before distribution.

## 2.3 Procedure

Ethical approval was obtained from the IRB in Jordan University of Science and Technology, number (34/132/2020). An information sheet was provided to the protentional participants. No consent was required as the participants were assured of data encryption and anonymity. Those who responded to the online invitation were considered consenting to participate in the study and were included in the study sample. Participant's anonymity was maintained, no name had to be written on the questionnaire, and only number codes were used. Moreover, participants were informed that their responses were completely confidential and were only be used for study purposes. For the use of the instrument, approval of the original author was sought.

## 2.4 Data analysis plan

Data analysis was done using IBM's SPSS version 23 (IBM, 2017). The descriptive data analysis included the means and standard deviations for continuous variables and frequencies and percentages for categorical variables. Pearson correlation coefficient was used to describe the direction, strength, and significance of the relationship between horizontal violence and intent to leave. An Independent t-test was used to compare horizontal violence results between public and private hospitals.

# 3 Result

## 3.1 Demographic characteristics of participants

Collection of data took place between March 31 and July 28, 2020. The study team sent 520 invitations and questionnaires to registered nurses and 436 participants responded and returned filled surveys. The response rate was 83.8%. The demographic variables include age, gender, marital status, years of experience, and educational level. Table 1 shows the breakdown of the characteristics of the sample.

## 3.2 Horizontal violence

All responses under the scale ranged from 1 to 5, according to the tool. Employees with an average of the sum score lower than 33 are not victims of horizontal violence, employees with

**Table 1. Demographic profile of the participants.**

| Category | Subcategory | Freq | % | Mean (SD) |
|---|---|---|---|---|
| Age | | | | 31.28 (7.33) |
| Age (category) | ≤ 25 years | 95 | 21.8% | |
| | 26 to 30 years | 141 | 32.3% | |
| | 31 to 35 years | 103 | 23.6% | |
| | 35 to 40 years | 44 | 10.1% | |
| | ≥41 years | 53 | 12.2% | |
| | **Total** | **436** | **100.0%** | |
| Gender | Male | 130 | 29.8% | |
| | Female | 306 | 70.2% | |
| | **Total** | **436** | **100.0%** | |
| Marital status | Single | 178 | 40.8% | |
| | Married | 238 | 54.6% | |
| | Divorced / Widow | 20 | 4.6% | |
| | **Total** | **436** | **100.0%** | |
| Educational level | Diploma | 82 | 18.8% | |
| | Bachelor | 277 | 63.5% | |
| | Master or PhD | 77 | 17.7% | |
| | **Total** | **436** | **100.0%** | |
| Sector | Governmental | 262 | 60.1% | |
| | Private | 174 | 39.9% | |
| | **Total** | **436** | **100.0%** | |
| Years of Experience (YOE) | | | | 8.40 (6.65) |
| YOE (category) | ≤1 year | 42 | 9.6% | |
| | 2–5 years | 156 | 35.8% | |
| | 6–10 years | 107 | 24.5% | |
| | 11–15 years | 62 | 14.2% | |
| | ≥16 years | 69 | 15.8% | |
| | **Total** | **436** | **100.0%** | |

a score between 33 and 45 may be considered as victims of horizontal violence occasionally, and employees who score above 45 can be considered to be regular victims of horizontal violence [13]. The overall average for all items under NAQ was 43.73, representing a moderate level of horizontal violence. In terms of nurses' distribution according to the level of horizontal violence, around one-third (n = 151, 34.6%) were victims of horizontal violence regularly. In contrast (n = 138, 31.7%) were victims to horizontal violence occasionally, and (n = 147, 33.7%) did not experience horizontal violence.

### 3.3 Anticipated turnover scale

Based on the findings, the overall average for the anticipated turnover scale subscale was 4.92, indicating a strong willingness to leave the job. All responses under the subscale ranged from 1 to 7. An average score of 3.5 and above means a strong willingness to leave the job, while below 3.5 indicates a weak willingness to leave job [16]. the highest scored item was item 5 "If I got another job offer tomorrow, I would give it serious consideration" (M = 5.59, STD = 1.96), indicating that nurses are seriously considering changing their current work environment, while the lowest scored item was item 7 "I've been in my position about as long as I want to (R)" (M = 3.02, STD = 1.97) indicating that nurses are not satisfied with staying in this job position for a long while. Table 2 describes all items of the ATS.

**Table 2. Descriptive for the anticipated turnover scale.**

| Item # | Anticipated turnover scale | X | STD | Rank |
|---|---|---|---|---|
| 1 | I plan to stay in my position for a while. (R) | 3.06 | 1.95 | 11 |
| 2 | I am quite sure I will leave my position in the foreseeable future | 4.71 | 2.12 | 5 |
| 3 | Deciding to stay or leave my position is not a critical issue for me at this point in time. | 5.27 | 1.90 | 2 |
| 4 | I know whether or not I'll be leaving this agency within a short time | 4.47 | 1.94 | 6 |
| 5 | If I got another job offer tomorrow, I would give it serious consideration | 5.59 | 1.96 | 1 |
| 6 | I have no intention of leaving my present position. (R) | 3.61 | 2.28 | 8 |
| 7 | I've been in my position about as long as I want to. (R) | 3.02 | 1.97 | 12 |
| 8 | I am certain I will be staying here a while. (R) | 3.26 | 1.98 | 10 |
| 9 | I don't have any specific idea how much longer I will stay | 5.22 | 1.80 | 3 |
| 10 | I plan to hang on to this job for a while. (R) | 3.58 | 2.16 | 9 |
| 11 | There are big doubts in my mind as to whether I will really stay in this agency | 4.78 | 2.09 | 4 |
| 12 | I plan to leave this position shortly | 4.39 | 2.17 | 7 |
| | **Overall** | **4.92** | **0.75** | |

## 3.4 Relationship between horizontal violence and intention to leave

The researchers used Pearson's r coefficient test of correlation to investigate the relationship between horizontal violence and intention to leave. Results showed a weak positive relationship between horizontal violence and anticipated turnover, r = 0.13, p<0.01. Horizontal violence level according to demographic variables.

## 3.5 Horizontal violence and demographic variables

T-test was used to assess the differences in nurses' responses to the negative acts' questionnaire according to the type of hospital where they work. The results showed that there is no significant difference in findings between nurses from governmental sector (M = 2.01, SD = 0.85) and those from private sector (M = 1.96, SD = 0.80), t = 0.548, p>0.05.

Also, a t-test was used to assess the differences in nurses' responses to the negative acts' questionnaire according to their gender. The results showed a significant difference in the level of horizontal violence between male nurses and female nurses. Female nurses (M = 2.04, SD = 0.87) scored significantly higher than male nurses (M = 1.87, SD = 0.73), t = -2.01, p<0.05.

Regarding marital status, a one-way Analysis of variance (ANOVA) was used to assess horizontal violence level differences according to marital status. The results showed no significant difference in horizontal violence level according to marital status, F = 0.17, p>0.05.

One-way ANOVA was used to assess horizontal violence level differences according to years of experience in nursing. The results showed no significant difference in horizontal violence level according to years of experience, F = 0.497, p>0.05.

Moreover, one-way ANOVA was also used to assess differences in horizontal violence level based on educational level. The results showed a significant difference in responses based on educational level, F = 3.16, p<0.05. The post-hoc analysis using Fisher's Least Significant Difference (LSD) showed that nurses who hold a diploma degree perceived a significantly higher level of horizontal violence as compared to those who hold a bachelor's degree (P<0.05).

## 3.6 Correlation between horizontal violence, intent to leave, and demographic variables

The correlation matrix (see Table 3) shows that significant relationships for horizontal violence as only intention to leave (r = -0.128, p<0.05). Other relationships related to horizontal violence or intent to leave are not significant.

**Table 3. Correlation between horizontal violence, intent to leave.**

|  |  | Age in years | Years of Experience | Years of Experience in the Department | NAQAvg | ATSAvg |
|---|---|---|---|---|---|---|
| Age in years | R | 1 |  |  |  |  |
|  | P |  |  |  |  |  |
| Years of Experience | R | .841** | 1 |  |  |  |
|  | P | .000 |  |  |  |  |
| Years of Experience in the Dept | R | .606** | .724** | 1 |  |  |
|  | P | .000 | .000 |  |  |  |
| NAQAvg | R | .009 | -.041 | -.018 | 1 |  |
|  | P | .852 | .394 | .715 |  |  |
| ATSAvg | R | -.091 | -.079 | -.089 | -.128** | 1 |
|  | P | .058 | .099 | .064 | .007 |  |

## 4. Discussion

The current study aimed to determine the current horizontal violence level among nurses in selected hospitals in Jordan. Findings showed that more than two-thirds of nurses experienced horizontal violence in one form or another. Nurses experienced a moderate level of horizontal violence, which approximated the prevalence and magnitude of horizontal violence in the country and in the region based on previous studies [4, 17]. While findings suggest that the prevalence of horizontal violence among the sampled nurses were not significantly different from the norm, the results of the study verify the presence of horizontal violence in the nursing work environment, which should not be the case, mainly when healthcare organizations aim to reduce factors that impact the physical and psychological health and well-being of the nursing workforce. Moreover, the similarity of the results with the norm seems to support the propositions of other studies that investigated the persistence of horizontal violence within healthcare organizations: horizontal violence exists and persists because horizontal violence has been integrated deeply within institutions to the extent that nurses have difficulty in distinguishing it from normal behavior as actions that should be prohibited and eliminated [18–20].

Results showed that nurses had a high intent to leave. However, it is important to note that intent to leave is a dynamic phenomenon, which means that it does not necessarily equate to leaving the job, and the intent to leave can change over time depending on prevailing personal, social, economic, and psychological circumstances of the individual toward the employing healthcare organization [21]. Therefore, while there might be high intention to leave at the point the study was conducted, nurse managers still have the opportunity to address the intention and look for strategies that can be implemented to reduce, if not eliminate, the intention and encourage the nurses to increase commitment and motivation to stay working in the organization.

After determining the prevalence and level of horizontal violence and the extent of the intention to leave, the study aimed to determine a significant relationship between the 2 variables. Correlational analysis showed a significant positive but weak relationship between horizontal violence and intention to leave. Though the relationship is weak and counterintuitive and contrary to literature stating that horizontal violence nurses experience within their workplaces is strongly associated with higher intention to leave their current job [10, 22]. However, the weak relationship between horizontal violence and intention to leave in the study does not mean that there is no relationship between the 2 variables at all in reality, since horizontal violence has been well-documented to negatively influence several nurse- and job-related outcomes [23, 24]. It may have only been the case that a stronger relationship was not detected in

the study owing to several factors that could have compounded the results including the possible concurrent organizational implementation of strategies to address horizontal violence. Moreover, the weak but significant relationship also signifies the possibility that other variables that were not included in the study aside from horizontal violence may have had a stronger relationship with intent to leave at the time the study was conducted. Investigating what other factors contributed to the observed level of intent to leave merits future research.

The current study also examines whether there were significant differences on the observed level of horizontal violence based on the sociodemographic characteristics of the sampled nurses. The lack of significant differences based on age, length of experience, and clinical setting (i.e., government or private hospital) echoes the results of other epidemiological studies on horizontal violence–that the phenomenon affects all nurses regardless of age, place of work, or length of clinical experience. Interestingly, results suggest that female nurses experience a higher level of horizontal violence than male nurses. While it is easy to attribute this result to possible gender inequalities in the workplace, for which the study did not measure, this merits future research as it is necessary to understand the role of sex and gender in horizontal violence and whether other factors mediate the occurrence of horizontal violence when directed to a particular sex or group of people identifying with a particular gender. Moreover, results showed a significant difference in the level of horizontal violence based on educational attainment. Such observed differences can be linked with the level of awareness and ability to recognize horizontal violence. This means if higher educational attainment provides a better understanding of the phenomenon, then less witnessed or experienced incidents of violence can be achieved. Nevertheless, to grasp a better understanding of the role played by education in the occurrence of horizontal violence, it is necessary in the future to investigate whether increasing the level of awareness and recognition of horizontal violence can effectively reduce or eliminate its occurrence.

## 5. Limitations

The study has some limitations. First, due to underreporting of incidents related to horizontal violence in hospitals, it was impossible to have sufficient data from the incident reporting system as a reliable source for objective data. Second, the study was cross-sectional in nature, which means that the variables were only measured in a snapshot of time. Both horizontal violence and intent to leave are dynamic and amenable to change. But this also presents an opportunity for nurse managers to measure horizontal violence and intent to leave regularly over time. Monitoring of the 2 variables can be performed in relation to the implementation of strategies that aim to address them. Third, horizontal violence and intent to leave were measured using self-report questionnaires. Though the psychometric properties of the questionnaires were established in other studies before use, it did not offer the researcher the ability to delve into the experiences of nurses. Future research can enhance and triangulate quantitative data collection with qualitative approaches to extract rich information about exactly how nurses experience horizontal violence and how such experiences influence their intent to leave the organization. Finally, future studies may include more hospitals from different settings with a larger sample size and utilizing other methods for data collection such as staff interviews and analysis of related incidents.

## 6. Conclusion

Horizontal violence can be in the form of physical or verbal abuse of other employees. In this study, it was found that horizontal violence in Jordanian organization is moderate, and it has a significant relationship with nurses' intention to quit. Therefore, there is an urgent need to

establish a nationwide and organizational 'zero tolerance' policy for horizontal violence to provide a more supportive work environment for nurses and improve their retention.

## 7. Implications for nursing management

Nurses need to be aware of the presence and prevalence of the horizontal violence phenomenon in the healthcare organizations and what to do if they were victims of violence from colleagues, including reporting this issue to their managers and learning about techniques to respond to them perpetrators of violence. Nursing managers have a vital role in ensuring and enforcing the implementation of the 'zero violence' policy in their department, which means that no reported incident of horizontal violence will be ignored. Each identified case of horizontal violence will be investigated and addressed appropriately. Understanding the negative impact on horizontal violence and its relationship with nurses' intent to leave, they need to work collaboratively with the management to prevent it from occurring and take disciplinary actions as needed in case someone violated the policy. They also have a role to support the victim staff and assure them that such acts are not acceptable, and their concern will be taken seriously and immediately to prevent it from reoccurring, which helps to improve nurses' satisfaction and retention.

## Acknowledgments

We are grateful to all the Jordanian nurses who participated in this study.

## Author Contributions

**Conceptualization:** Ahmad H. Al-Nawafleh.

**Data curation:** Ahmad H. Al-Nawafleh, Zaid M. Al-Hamdan, Hala Bawadi.

**Formal analysis:** Zaid M. Al-Hamdan, Hamzah Bawayzah.

**Investigation:** Hamzah Bawayzah.

**Methodology:** Ahmad H. Al-Nawafleh, Zaid M. Al-Hamdan, Hamzah Bawayzah.

**Supervision:** Ahmad H. Al-Nawafleh, Zaid M. Al-Hamdan.

**Validation:** Hala Bawadi.

**Writing – original draft:** Hamzah Bawayzah.

**Writing – review & editing:** Ahmad H. Al-Nawafleh, Zaid M. Al-Hamdan, Hala Bawadi.

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
