## [Decision Letter · Decision Letter 0]

3 May 2024

PONE-D-24-04986The Influence of Horizontal Violence on intention to leave among Jordanian Nurses: A cross-sectional studyPLOS ONE

Dear Dr. Al-Nawafleh,

Thank you for submitting your manuscript to PLOS ONE. After careful consideration, we feel that it has merit but does not fully meet PLOS ONE’s publication criteria as it currently stands. Therefore, we invite you to submit a revised version of the manuscript that addresses the points raised during the review process.

We look forward to receiving your revised manuscript.

Kind regards,

Omar Mohammad Ali Khraisat, Associate Professor

Academic Editor

PLOS ONE

Journal Requirements:

Reviewers' comments:

Reviewer's Responses to Questions

**Comments to the Author**

1. Is the manuscript technically sound, and do the data support the conclusions?

Reviewer #1: Yes

2. Has the statistical analysis been performed appropriately and rigorously? 

Reviewer #1: Yes

3. Have the authors made all data underlying the findings in their manuscript fully available?

Reviewer #1: Yes

4. Is the manuscript presented in an intelligible fashion and written in standard English?

Reviewer #1: Yes

5. Review Comments to the Author

Reviewer #1: Completing the survey was deemed as consenting to be involved in the study. Respondents details were collated without identification of individuals. The study topic is globally recognized as impacting workplace moral, attrition and retention of staff. The impact of horizontal violence on healthcare environments in which nurses are employed is acknowledged and has been discussed in literature in many nations. This paper according to the authors is the first in Jordan to explore nurses experiences of horizontal violence and the intention to leave. Elevating workplace cultures as a human resource issue is necessary to ensure policies and practices are entrenched in all organisations that recognize and uphold positive workplace environments. I congratulate the authors on raising awareness of this topic through their research.

6. PLOS authors have the option to publish the peer review history of their article (what does this mean?). If published, this will include your full peer review and any attached files.

Reviewer #1: **Yes: **Professor Karen L Francis

School of Nursing, Paramedicine & Healthcare Science

Faculty of Science

Charles Sturt University

Wagga Wagga NSW Australia

---

## [Author Response · Author response to Decision Letter 0]

13 May 2024

Dear Reviewer(s),

Thank you for your time and effort of reading and commenting on our manuscript. We would like also to thank you for the supportive message that you written in the comments about the topic and its importance to conduct research. 

All your comments were carefully considered and amended on the track changes file and then accepted on the manuscript file too. 

We hope this will be satisfactory and will take our submission to the next step. 

Regards

---

## [Decision Letter · Decision Letter 1]

11 Jul 2024

The Influence of Horizontal Violence on intention to leave among Jordanian Nurses: A cross-sectional study

PONE-D-24-04986R1

Dear Dr.,

We’re pleased to inform you that your manuscript has been judged scientifically suitable for publication and will be formally accepted for publication once it meets all outstanding technical requirements.

Kind regards,

Omar Mohammad Ali Khraisat, Associate Professor

Academic Editor

PLOS ONE

Additional Editor Comments (optional):

Reviewers' comments:

Reviewer's Responses to Questions

**Comments to the Author**

1. If the authors have adequately addressed your comments raised in a previous round of review and you feel that this manuscript is now acceptable for publication, you may indicate that here to bypass the “Comments to the Author” section, enter your conflict of interest statement in the “Confidential to Editor” section, and submit your "Accept" recommendation.

Reviewer #2: All comments have been addressed

2. Is the manuscript technically sound, and do the data support the conclusions?

Reviewer #2: Yes

3. Has the statistical analysis been performed appropriately and rigorously? 

Reviewer #2: Yes

4. Have the authors made all data underlying the findings in their manuscript fully available?

Reviewer #2: Yes

5. Is the manuscript presented in an intelligible fashion and written in standard English?

Reviewer #2: (No Response)

6. Review Comments to the Author

Reviewer #2: (No Response)

7. PLOS authors have the option to publish the peer review history of their article (what does this mean?). If published, this will include your full peer review and any attached files.

Reviewer #2: No

---

## [Editor Report · Acceptance letter]

10 Sep 2024

PONE-D-24-04986R1 

PLOS ONE

Dear Dr. Al-Nawafleh, 

I'm pleased to inform you that your manuscript has been deemed suitable for publication in PLOS ONE. Congratulations! Your manuscript is now being handed over to our production team.

Kind regards, 

on behalf of

Dr. Omar Mohammad Ali Khraisat 

Academic Editor

PLOS ONE